

# Genomic analysis of variability in Delta-toxin levels between *Staphylococcus aureus* strains

Michelle Su[1], James T. Lyles[2], Robert A. Petit III[1], Jessica Peterson[1], Michelle Hargita[1], Huaqiao Tang[2], Claudia Solis-Lemus[3], Cassandra L. Quave[2,4] and Timothy D. Read[1,3]

[1] Division of Infectious Diseases, Department of Medicine, School of Medicine, Emory University, Atlanta, GA, United States of America
[2] Center for the Study of Human Health, College of Arts and Sciences, Emory University, Atlanta, GA, United States of America
[3] Department of Human Genetics, School of Medicine, Emory University, Atlanta, GA, United States of America
[4] Department of Dermatology, School of Medicine, Emory University, Atlanta, GA, United States of America

Corresponding author
Timothy D. Read, tread@emory.edu

## ABSTRACT

**Background**. The delta-toxin (δ-toxin) of *Staphylococcus aureus* is the only hemolysin shown to cause mast cell degranulation and is linked to atopic dermatitis, a chronic inflammatory skin disease. We sought to characterize variation in δ-toxin production across *S. aureus* strains and identify genetic loci potentially associated with differences between strains.

**Methods**. A set of 124 *S. aureus* strains was genome-sequenced and δ-toxin levels in stationary phase supernatants determined by high performance liquid chromatography (HPLC). SNPs and kmers were associated with differences in toxin production using four genome-wide association study (GWAS) methods. Transposon mutations in candidate genes were tested for their δ-toxin levels. We constructed XGBoost models to predict toxin production based on genetic loci discovered to be potentially associated with the phenotype.

**Results**. The *S. aureus* strain set encompassed 40 sequence types (STs) in 23 clonal complexes (CCs). δ-toxin production ranged from barely detectable levels to >90,000 units, with a median of >8,000 units. CC30 had significantly lower levels of toxin production than average while CC45 and CC121 were higher. MSSA (methicillin sensitive) strains had higher δ-toxin production than MRSA (methicillin resistant) strains. Through multiple GWAS approaches, 45 genes were found to be potentially associated with toxicity. Machine learning models using loci discovered through GWAS as features were able to predict δ-toxin production (as a high/low binary phenotype) with a precision of .875 and specificity of .990 but recall of .333. We discovered that mutants in the *carA* gene, encoding the small chain of carbamoyl phosphate synthase, completely abolished toxin production and toxicity in *Caenorhabditis elegans*.

**Conclusions**. The amount of stationary phase production of the toxin is a strain-specific phenotype likely affected by a complex interaction of number of genes with different levels of effect. We discovered new candidate genes that potentially play a role in modulating production. We report for the first time that the product of the *carA* gene is necessary for δ-toxin production in USA300. This work lays a foundation for future

work on understanding toxin regulation in *S. aureus* and prediction of phenotypes from genomic sequences.

# INTRODUCTION

*Staphylococcus aureus* is a common causative agent of nosocomial and community-acquired infections, encoding a wide variety of factors that damage the host and evade immunity. Central to its ability to cause disease is its large repertoire of toxins. *S. aureus* can produce at least 13 extracellular toxins (*Grumann, Nübel & Bröker, 2014*; *Otto, 2014*; *Laabei et al., 2015*), including phenol-soluble modulins (PSMs) (*Peschel & Otto, 2013*), alpha-toxin (*Bhakdi & Tranum-Jensen, 1991*), Panton-Valentine Leukocidin (PVL) (*Genestier et al., 2005*), and δ-toxin (*Wang et al., 2007*).

Toxin expression levels are subject to evolutionary trade-offs between survival and transmission in different environments (*Laabei et al., 2015*; *Young et al., 2017*). Toxins contribute to important biological functions: In *S. aureus*, alpha-toxin is important for initial cell-to-cell contacts in biofilm formation, beta-toxin contributes to biofilm structure and growth via crosslinking, and PSMs are involved in detachment of cells for dispersal (*Rudkin et al., 2017*). In addition, expression of toxins is essential to skin and soft tissue infections and other common diseases caused by the bacterium (*Xu & McCormick, 2012*; *Otto, 2013*; *Peschel & Otto, 2013*; *Kitur et al., 2015*). However, during chronic *S. aureus* infections, toxin production is a contra-indication of disease as reduced toxicity mutants may have situationally increased fitness (*Cheung et al., 2014*; *Soong et al., 2015*; *Rose et al., 2015*; *Laabei et al., 2015*). Dysfunction in the Agr quorum sensing system (*Novick, 2003*), central to upregulation of many toxins, has been linked to longer durations of bacteremia (*Fowler Jr et al., 2004*; *Sakoulas et al., 2005*). Similarly, mutational inactivation of another regulator, Rsp, which promotes *S. aureus* infection and virulence (*Li et al., 2015*), allows for prolonged survival in chronic infections (*Das et al., 2016*).

In this study, we focus on the genetics of strain-specific differences of δ-toxin expression. δ-toxin is an amphipathic peptide in the PSM family. It can form pores on the surface of host cells, eliciting a pro-inflammatory response or cytolysis at high concentrations (*Bernheimer & Rudy, 1986*; *Kasimir et al., 1990*; *Otto, 2014*). δ-toxin is the product of the *hld* gene, which is part of the Agr quorum sensing system. The Agr operon consists of two divergently transcribed operons P2 and P3. The P2 operon encodes the four genes necessary for quorum sensing and activates the P3 operon, which transcribes the main effector of the Agr system, a 514-nucleotide regulatory molecule RNAIII. RNAIII also contains the *hld* gene encoding the 26 amino acid δ-toxin peptide, which has been found only in *S. aureus* and *S. epidermidis* (*McKevitt et al., 1990*). In a community-associated MRSA (CA-MRSA) bacteremia mouse model, PSM $\alpha$ and δ-toxin were shown to be important for disease severity, indicating their importance as virulence factors (*Wang et al., 2007*; *Peschel &*

*Otto, 2013*). However, δ-toxin is the only PSM shown to induce mast cell degranulation (*Nakamura et al., 2013*) and increase the severity of *S. aureus* mediated Atopic Dermatitis (AD), a chronic inflammatory skin disease, affecting 15–30% of children and 5% of adults in the US and industrialized countries (*Williams & Flohr, 2006*; *Pustišek, VurnekŽivković & Šitum, 2016*). Despite its importance, we know little of the natural variation in production of the δ-toxin molecule between *S. aureus* strains and the genetic factors that influence this trait. Therefore, we queried the range of δ-toxin production in a diverse set of *S. aureus* strains and attempted to determine if there are genetic loci strongly associated with δ-toxin production by using bacterial genome-wide association study (GWAS) methods. We then analyzed the performance of identified genome variants and metadata for predicting δ-toxin production.

## MATERIALS & METHODS

### Strains and growth conditions

Network on Antimicrobial Resistance in *Staphylococcus aureus* (NARSA) and Nebraska Transposon Mutant Library (NTML) strains were acquired from BEI resources (https://www.beiresources.org/) (Table S1).  For δ-toxin assays, bacteria were grown on tryptic soy agar (TSA) plates overnight (18–24 hours) at 37 °C.  TSA plates used for NTML strains had the addition of erythromycin (5 μg/ml). Cultures from a single colony were inoculated and grown overnight in tryptic soy broth (TSB) at 37 °C, a 45° angle, and 200 rpm.  Final cultures were standardized to a starting cell density of $5 \times 10^5$ CFU/ml of TSB and grown for 15 hours at 37 °C, a 45° angle, and 275 rpm (*Quave & Horswill, 2018*).

### Whole-genome shotgun sequencing

DNA extraction and paired-end library prep were performed as manufacturer's instructions (Wizard Genomic DNA Purification Kit, Promega; Nextera XT DNA Library Prep Kit, Illumina).  Genome sequencing was performed using both Illumina HiSeq and MiSeq. Raw read data were deposited in the NCBI Short Read Archive under project accession number PRJNA289526. 102/124 strains had more than 40x average genome coverage, and the minimum coverage of any strain was 33x.

### Genome assembly and annotation

Genomes were processed using the Staphopia pipeline (*Petit & Read, 2018*). BBduk (v37.66) (*Bushnell, 2016*) was used to eliminate Illumina adapters, trim low quality ends (base quality <20), and filter out low quality reads (mean read PHRED quality <20).  Read error correction and *de novo* genome assembly was performed using SPAdes (v3.11.11) (*Bankevich et al., 2012*). Genome assemblies were annotated with Prokka (*Seemann, 2014*) (v1.12) using its default database. SNP-sites was used to call single nucleotide polymorphisms (SNPs) in the core genome alignment with *S. aureus* N315 as the reference strain (*Page et al., 2016*). Agr type was determined using BLAST to query genome assemblies for the *agrD* nucleotide sequences of defined agr types:  I (AB492152.1), II (AF001782), III (AF001783), and IV (AF288215).  For all but four samples, 100% coverage and >95% identity were used to identify Agr type. NRS168, NRS182 NRS235, and NRS260 Agr types

were determined based on available metadata. Untyped strains (NA) returned no BLAST results likely due to contig boundaries falling in this region. MLST (multilocus sequencing type) was determined using the SRST2 tool (*Inouye et al., 2014*) with the PubMLST database (*Jolley, Bray & Maiden, 2018*).

## Phylogenetic tree estimation

A core genome alignment of 999,473 base-pairs (bp) from the 124 NARSA strains was obtained from Roary (*Page et al., 2015*; *Tange, 2011*) (v3.11.2). Gubbins was used to remove potential recombination regions and to obtain a downsized core genome alignment of 42,406 bp containing only polymorphic sites (*Croucher et al., 2015*). A final maximum likelihood (ML) tree was obtained with RAxML (v8.2.10) with 100 bootstraps and a GTRGAMMA model (*Stamatakis, 2014*).

## Toxin identification using HPLC

High performance liquid chromatography (HPLC) methods were employed to detect and quantify the levels of δ-toxin present in supernatants of 124 NARSA strains following established procedures (*Quave & Horswill, 2018*). Briefly, a 1.5 ml *S. aureus* culture grown for 15 hours as described above was centrifuged and the supernatant transferred to an HPLC vial and frozen at −20 °C until ready for HPLC testing. HPLC was performed with the following parameters: 500 μl injection, flow rate of 2 mL/min, and UV/Vis monitored at 214 nm using solvents (a) 0.1% (vol/vol) trifluoracetic acid in water and (b) 0.1% trifluoracetic acid in acetonitrile. Peaks at retention time ∼7.2 min and ∼7.5 min corresponding to deformylated and formylated δ-toxin respectively, were quantified by taking the sum of the total peak area. Peak areas were normalized using $OD_{600}$ readings of the cultures. Prior studies using this HPLC method confirmed peak identity at these retention by LC-MS (*Somerville et al., 2003*; *Quave, Plano & Bennett, 2011*). Analyses were performed on three replicate supernatants per strain.

For subsequent analyses treating toxin production as a continuous variable, we used a Box-Cox power transformation to achieve a more symmetric distribution and thus satisfy the normality assumption of phylogenetic regression and other comparative models. For analyses treating toxin production as a binary Low/High, we used a cutoff of 20,000, which clusters strains on the left-side of the distribution and split the data into 109 low and 15 high toxin producers.

## Hemolysis assay

TSA with 5% rabbit's blood was used to test transposon strains for reduced hemolysis as rabbit blood is more susceptible to δ-toxin. TSA II with 5% sheep's blood agar was used to test the hemolysis profile of complemented strains. Strains were spotted and incubated at 37 °C for 24 hours before incubation at 4 °C for an additional 24 hours. Photos of the plates were taken with the use of a lightbox to illuminate hemolysis zones. Images were imported into software ImageJ (*Schneider, Rasband & Eliceiri, 2012*) to increase the contrast of the image and for measurement of hemolysis zones by taking the hemolysis measurement and subtracting colony size.

## GWAS

All GWAS analyses were done with 106 *S. aureus* strains. NRS168, NRS252-NRS256, NRS259, NRS260, NRS262, NRS264-NRS266, NRS271, NRS272, NRS275, NRS383, NRS386, NRS387, and NRS408 were later phenotyped and included in all other analyses.

SEER (*Lees et al., 2016*) (v 1.1.4) was run according to https://github.com/johnlees/seer/wiki/Tutorial. In brief, kmers used for SEER were counted using fsm-lite (https://github.com/nvalimak/fsm-lite) using genome assemblies in fasta format as input. Population structure was estimated using Mash (*Ondov et al., 2016*) to sketch assemblies and output pairwise distances between all samples. SEER scripts were used to create a distance matrix. Six dimensions was chosen based on scree plot output (Fig. S1). SEER was run using a binary phenotype with *p* value filtering off. QQ plots were made in R to ensure that population structure was properly accounted for. A minor allele frequency (MAF) of .20 was chosen as regression analysis with kmers of lower MAF tend to fail or have high standard errors. Significant kmers were kmers with likelihood ratio test *p*-values lower and equal to the Bonferroni correction of .05/n, where n is the number of kmers tested. Significant kmers were mapped to reference genome N315 (NC_002745.2) using BLAST (*Camacho et al., 2009*) optimized for short queries. Bedtools (v2.27.1) (*Quinlan & Hall, 2010*) was then used to annotate the matches.

For treeWAS (*Collins & Didelot, 2017*), a binary core SNPs matrix was generated from snp-sites output. To account for the right skewed distribution of δ-toxin production, the values were transformed into ranks. treeWAS was run with 3 unrooted trees (NARSA strains alone, NARSA strains plus ST93, NARSA strains plus *S. argenteus*) generated from RAxML (v8.2.10) to limit false positives generated from an incorrect phylogeny, and the intersection of all loci identified was considered significant.

For bugwas (*Earle et al., 2016*), we used a modified version of GEMMA 0.93 (*Zhou & Stephens, 2012*) with a centered relatedness matrix (GEMMA option -gk 1) created using BIMBAM files and a binary toxin phenotype file and set Minor Allele Frequency of 0. Biallelic core SNPs were used to create a mean genotype file, and SNP positions were noted in a SNP annotation file with the chromosome number set to 24 to indicate one allele. A nucleotide matrix of core SNPs, a binary phenotype file, and an unrooted phylogenetic tree created by RAxML (v8.2.10) were used to run the bugwas R package. A Bonferroni correction of .05/n was used, where n was the sum of phylogenetic patterns represented by the bi and tri allelic SNPs.

DBGWAS (*Jaillard et al., 2018*) (v0.5.0) was run using genome assemblies in fasta format, a binary phenotype, unrooted phylogenetic tree created by RAxML (v8.2.10), and DBGWAS resistance and UniProt databases for annotation. A false discovery rate (FDR) of 5% was used to determine significant kmers.

## Phylogenetic regression

We fit three phylogenetic comparative models: (1) a phylogenetic regression to study the association between δ-toxin production and several covariates, (2) a Pagel's lambda model to estimate the phylogenetic signal of δ-toxin production, and (3) an ancestral

state reconstruction of δ-toxin production along the branches of the *Staphylococcus aureus* phylogeny.

For the phylogenetic regression model, we included δ-toxin level as a continuous response for 106 strains and 11 predictors: clonal complex (CC), methicillin-resistant *S. aureus* (MRSA), Agr type (agr), and variants associated with δ-toxin production in *S. aureus* identified using SEER (*Lees et al., 2016*) and bugwas (*Earle et al., 2016*) and DBGWAS (*Jaillard et al., 2018*) (Table S3). *isdC* and WP_000894032.1 were excluded from the analysis due to large standard errors while sequence type (ST) was excluded due to similarity to CC. Bugwas variants were represented in the analysis as 6 phylogenetic patterns. We used the julia package PhyloNetworks (*Solís-Lemus, Bastide & Ané, 2017*; *Bastide et al., 2018*) to fit the phylogenetic regression, to estimate the phylogenetic signal in δ-toxin through Pagel's λ transformation, and to reconstruct the ancestral states. To tease apart which if any of these factors truly impact δ-toxin production, we performed a phylogenetic regression. First, with Pagel's lambda model, we estimated a strong phylogenetic signal ($\lambda = .504035$) for the δ-toxin production using a rooted (by *S. argenteus*) phylogenetic tree calibrated to be consistent with time. This estimate did not entirely fit under the Brownian Model (BM) assumption, which requires $\lambda \approx 1$. Regardless of this, we assumed a BM for the evolution of δ-toxin production in the phylogenetic regression model. The rationale for the use of BM was its simplicity as well as the shown robustness to model misspecification (*Bastide et al., 2018*).

### Extreme Gradient Boosted Tree Classifier

The R package xgboost (*Chen & Guestrin, 2016*) was used to create predictive classifiers with strain metadata and genetic features from the GWAS. The predictor was trained using stratified 10-fold cross-validation wherein 90% is used for training and 10% is used for validation. Model performance metrics such Area Under Receiver Operating Characteric (AUROC) and Cohen's Kappa were calculated using R packages pROC and irr respectively.

### Other statistical analysis

Association of MRSA/MSSA status, Agr type, and CC to toxin production was performed with Kruskall-Wallis and pairwise Mann-Whitney U tests using continuous δ-toxin as the dependent variable. A Bonferroni correction was applied to test *p*-values to account for multiple tests. Pagel's lambda and Blomberg's K (*Pagel, 1999*; *Blomberg, Garland Jr & Ives, 2003*) were estimated using R package phytools using an unrooted phylogenetic tree obtained from RAxML and no calibration to the branch lengths. All analyses were performed using the R (*R Core Team, 2016*) and Julia programming language (*Bezanson et al., 2017*) for statistical computing.

### Phenotypic analysis of toxin phenotypes of transposon mutant strains

Nine transposon mutants in genes potentially associated with δ-toxin production were selected from the USA300 Nebraska transposon library (*Fey et al., 2013*). In addition, we selected an *agrA* mutant as a positive control for δ-toxin disruption and one randomly chosen mutant with no known association from the GWAS experiments as a negative control. The gene disrupted was a phi77 ORF109-like protein, SAUSA300_1928,

WP_000582165.1. All transposon mutants were transduced into an isogenic USA300 JE2 background and confirmed by PCR (Table S2). HPLC assays and hemolysis assays for δ-toxin were as previously described. Complementation was performed by cloning PCR fragments containing the USA300 genes into the pOS1-Plgt vector using splicing overlap extension PCR (*Bubeck Wardenburg, Patel & Schneewind, 2007*). In brief, the plasmid and genes were PCR amplified to contain complementary overhangs. The purified products were then mixed and subject to another round of PCR with no primers. This reaction was used to transform IM08B *E. coli* (*Monk et al., 2015*). The plasmid was purified and electroporated into the mutant strains as previously described (*Monk et al., 2012*). A pOS1-Plgt only plasmid was used as a control for complementation experiments. The *C. elegans* virulence assays were performed using *C. elegans* strain N2. Nematode population synchronization was performed as in *Penley & Morran (2018)*. Populations were bleached in 20% household bleach and M9 buffer and plated on OP50 until L4 larval stage (48 hours at 20 °C). Worms were subsequently washed off, counted, and ∼200 were plated on control OP50 plates and *S. aureus* lawns on BHI agar. *S. aureus* plates were prepared 24 hours prior by adding 200 ul of an overnight culture and growing at 24 °C. At 24 hours and 48 hours, plates were scored by counting live worms. Worm counts on OP50 plates were used to normalize mortality calculations and to account for plating efficiency.

## RESULTS

### δ-toxin production level is highly variable between *S. aureus* strains and is associated with MSSA/MRSA and Clonal Complex

We used high performance liquid chromatography (HPLC) to quantify stationary phase δ-toxin production in 124 publically available *S. aureus* strains from the Network on Antimicrobial Resistance in *Staphylococcus aureus* (NARSA) collection, which represents diverse taxonomic groups within the species (Table S1). The strains, which were shotgun sequenced using Illumina technology, were a diverse representation of the *S. aureus* species, consisting of 40 sequence types (STs) in 23 clonal complexes (CCs). There was considerable variation in the total δ-toxin production (sum of the formylated and deformylated δ-toxin peptides) between strains (Fig. 1, Fig. S2). The distribution most closely fits a gamma model with a strong left skew. Production ranged from zero to 97,235 units, with a median value of 8,295. The majority of strains produced less than 10,000 units; 118 of the 124 strains (95%) produced less than 30,000.

When toxin production was mapped onto the strain phylogeny, it was apparent that there was variation in the average level between clonal complexes even though there was also a large variation in the phenotype within CCs (Fig. 2). Two tests for phylogenetic signal of the trait, Blomberg's K and Pagel's lambda (*Pagel, 1999*; *Blomberg, Garland Jr & Ives, 2003*), returned statistically robust scores (K = .019, $p = .016$/ λ = .99, $p = 1.55e-48$; for both measures, a value of 1 indicates trait similarity measured by variance (K) or correlation (λ) as expected under Brownian evolution). A Pagel's lambda value of ∼1 indicates a strong phylogenetic signal, while the low Blomberg's K indicates that the variance that exists within δ-toxin production is primarily on the tips of the trees within clades and does not wholly fit

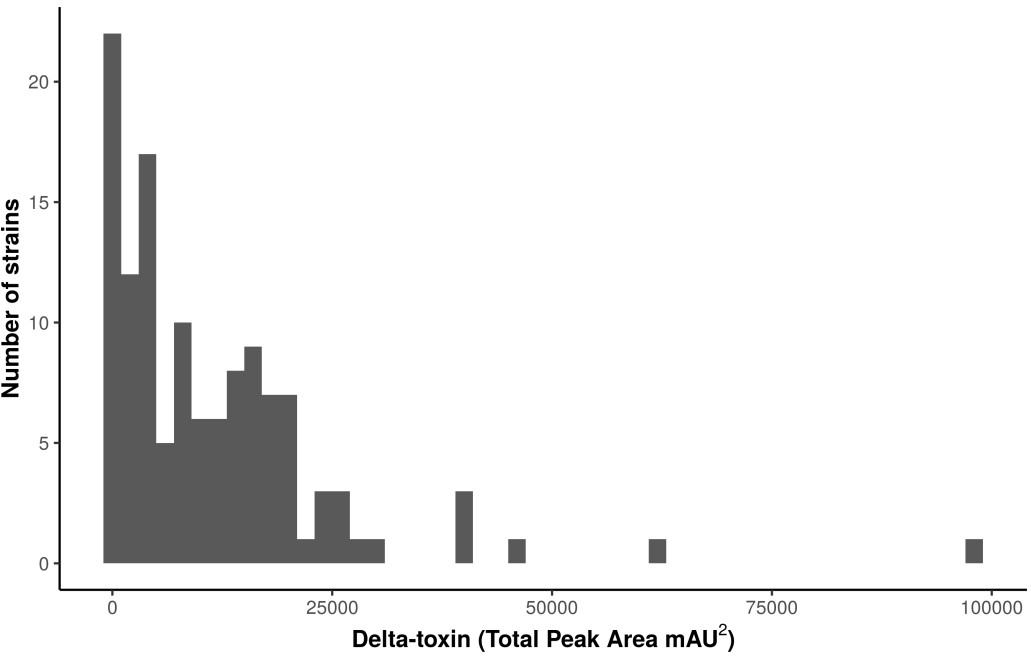

**Figure 1** **Characterization of δ-toxin production in 124 _S. aureus_ strains.** Supernatants from 124 _S. aureus_ strains from the Network on Antimicrobial Resistance in Staphylococcus aureus (NARSA) repository were subjected to high performance liquid chromatography (HPLC). The amount of δ-toxin present in samples is proportional to the area under the peak generated by UV absorbance when passing through the HPLC detector. Production ranged from 0 to 97,234 mAU$^2$.

a Brownian model of evolution for δ-toxin. When variation between clades was analyzed pair-wise using Kruskal-Wallis and Mann-Whitney tests, CC30 (Average δ-toxin 4299) was found to have significantly lower δ-toxin production than CC45 (Average δ-toxin 30955, $p = .027$) and CC121 (Average δ-toxin 17693, $p = .00042$, Fig. 3C). Ancestral reconstruction of the δ-toxin phenotype (Fig. S3) suggested that high δ-toxin producing clades such as CC45, CC890 and CC72 had arisen independently in the _S. aureus_ species from a low producing ancestor.

Agr groups have been suggested to be associated with differences in _S. aureus_ cytotoxicity (_Jarraud et al., 2002_; _Collins, Buckling & Massey, 2008_). All four Agr groups were present in our samples (I: 50, II: 22, III: 38, IV: 11, NA: 3). We found significant differences in toxin production between Agr I and III ($p = .022$) using Kruskal-Wallis test and between Agr III and IV ($p = .00049$) using pairwise Mann-Whitney U tests. Agr I and IV have higher mean levels than Agr II and III (Fig. 3B). Methicillin resistance has also been previously indicated to interfere with the Agr quorum sensing system and thus toxin production (_Rudkin et al., 2012_). Within our set of strains, MRSA strains were found to have lower δ-toxin than MSSA strains by Mann-Whitney U ($p = 0.024$, Fig. 3A). Some caution must be used when assigning causality as clonal complex, Agr group and MRSA status are strongly confounded, but we also found that MRSA status and Agr type III were significant negative predictors of δ-toxin level in a phylogenetic regression (see Methods).
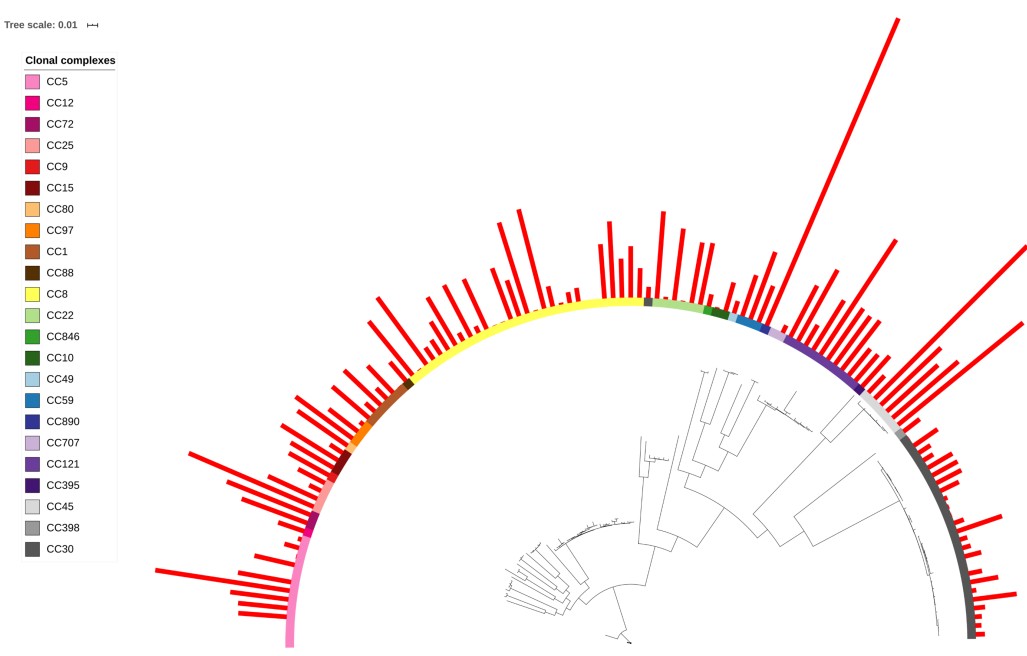

**Figure 2** **δ-toxin production across the *S. aureus* phylogeny.** A total of 124 *S. aureus* strains in 23 clonal complexes (CC) were used to create a core genome phylogeny using RAxML. Clonal complexes are represented in the inner ring and color coded. Red bars represent δ-toxin values from HPLC. Missing bars indicate δ-toxin was undetectable by HPLC.

## Diverse genetic loci are associated with variation in individual strain δ-toxin production levels

To ascertain individual genomic variants that may potentially be associated with the differences in toxin production, we used four recently published programs for bacterial GWAS (genome-wide association studies). The programs differ in the population structure correction, the types variants tested and whether continuous or binary phenotypes could be tested. SEER (*Lees et al., 2016*) is an alignment-free method that uses kmers as features to create a distance matrix and a fixed effects model to correct for population structure. SEER therefore allows discovery of both core and accessory gene variants associated with the phenotype. For the purposes of GWAS, we defined the binary toxin phenotype at a cutoff of 20,000 units (Fig. 1), which gave a set of 87 strains in the "low" toxin and 19 in the high toxin category. This threshold was chosen to separate the very high-producing strains from the main mass (Fig. 1). Using the binary phenotype, SEER identified three genes having more than ten kmers with statistically significant association (Table S3). Polymorphisms in *isdC*, *glpD*, and a gene encoding YbbR-like domain-containing protein (WP000894032.1) were found to be negatively associated with δ-toxin production. Bugwas (*Earle et al., 2016*) is an alternative distance-based GWAS program that uses principal components as random effects for population structure control. Bugwas analysis, performed with a binary phenotype, produced six phylogenetic patterns of SNPs (Table S3). DBGWAS (*Jaillard et al., 2018*) is a kmer based alignment-free method which relies on De Brujin graphs to interpret genomic variation but uses bugwas for downstream analyses. Running DBGWAS

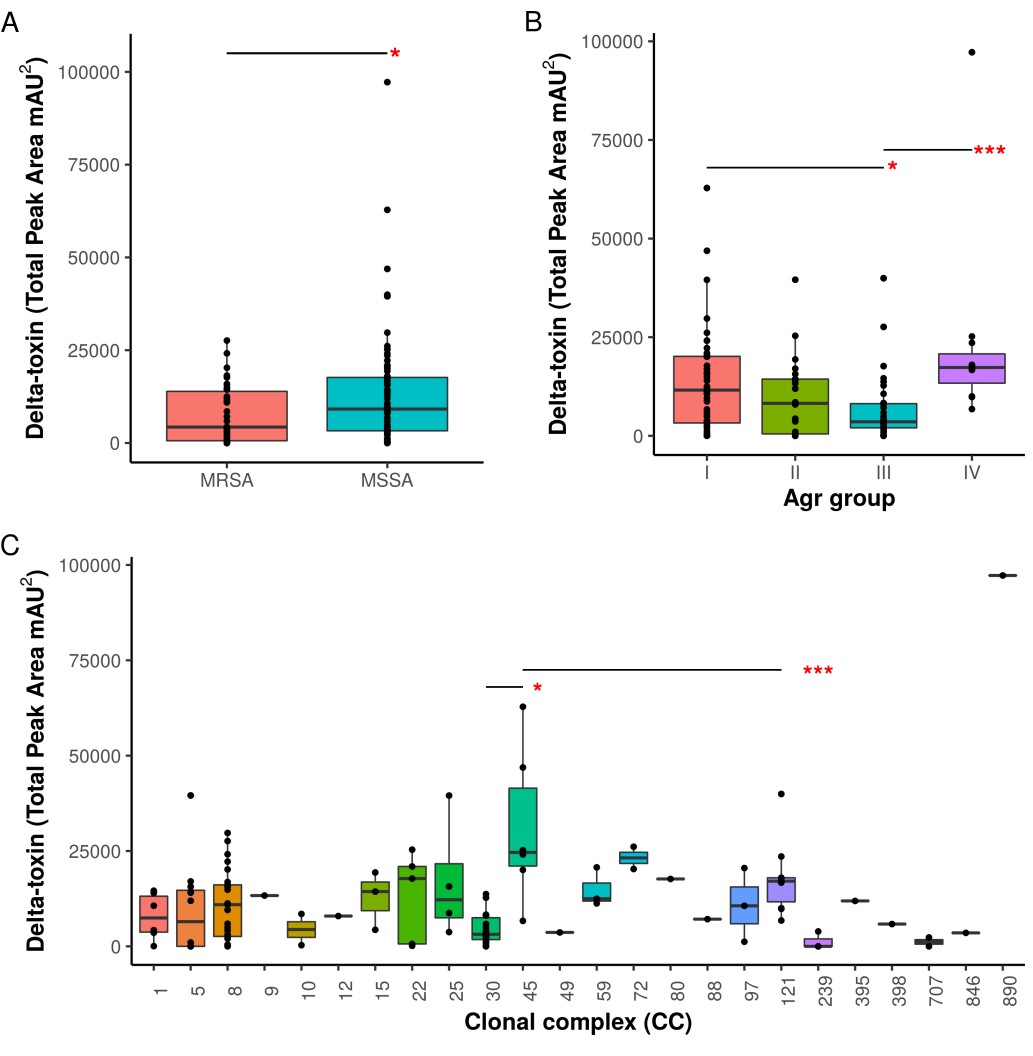

**Figure 3** **Associations of δ-toxin production to methicillin resistance, Agr type and clonal complex.**
(A) Differences in δ-toxin production between MSSA and MRSA strains. (B) Differences in δ-toxin production between agr types. (C) Differences in δ-toxin production between CCs. $*p < .05$, $**p < .01$, $*** < .001$.

with a binary phenotype yielded one hit in an intergenic region near staphopain A (Table S3). treeWAS (*Collins & Didelot, 2017*) differed from the other three programs in using the strain phylogeny to model changes associated with phenotype changes at the tip and within the structure of the tree. Using a phylogenetic tree can be more accurate than distance estimation if the tree is correct, therefore for robustness we used three separate trees (NARSA strains only, NARSA strains with a ST93 outgroup, NARSA strains with an *S. argenteus* outgroup) and pooled common loci. Using a ranked phenotype, genes common to all three analyses were *fadD, vraD, degA, gdpP, ggt, sufB, opcR, rebM, thiD,* and three uncharacterized proteins (Table S3).

When the results from the four approaches were aggregated (Table S3), we noted the majority of the genetic loci were in genes encoding enzymes that were part of conserved

**Table 1  XGBoost models were trained with and without metadata (ST, CC, Agr group, and MSSA/MRSA status).** Ten-fold cross-validation was used to assess model performance. For all measures, the average performance across cross-validation is reported. For non-binary classification, individual precision and recall measures are weighted according to its proportion in the overall dataset during averaging, and Cohen's Kappa is calculated using squared weights. Specificity is not measured for non-binary classification models as it is included in the weighted precision and recall measures.

| Model | Precision | Recall | Specificity | AUROC (95% CI) | Cohen's Kappa ($p$-value) |
|---|---|---|---|---|---|
| Binary predictor +/ − metadata | .875 | .333 | .990 | .697 (.553, .840) | .429 (.000000037) |
| Four category predictor + metadata | .423 (weighted) | .443 (weighted) | N/A | .664 (.597, .731) | .255 (.00441) (weighted) |
| Four category predictor − metadata | .451 (weighted) | .326 (weighted) | N/A | .667 (.576, .758) | .133 (.131) (weighted) |

metabolic pathways. Forty-two variants were synonymous mutations, and only nine were non-synonymous (6 loci from the bugwas analysis and 3 from treeWAS). None of the four GWAS approaches found any significant genetic loci in common.

We attempted to determine if machine learning approaches could predict δ-toxin from genome sequences by integrating information from the diverse GWAS analyses with MRSA/MSSA, Agr group and genotype (CC or ST). We chose Extreme Gradient Boosting (XGBoost) (*Friedman, 2001*; *Chen & Guestrin, 2016*) which uses decision tree ensembles to predict from the given set of features. XGBoost has been used to successfully predict biologically relevant phenotypes such as antibiotic resistance in *Enterobactericiae* and *Salmonella* (*Nguyen et al., 2018*; *Nguyen et al., 2019*) as well as RNA-protein interactions (*Jain, Gupte & Aduri, 2018*), protein-protein interactions (*Wang, Liu & Deng, 2018*; *Sanchez-Garcia et al., 2019*), and RNA methylation (*Qiang et al., 2018*). An XGBoost model was trained with stratified 10-fold cross-validation. When using a binary δ-toxin phenotype (>20000), the model had a precision of .875 and a recall of .333. Specificity was .990, and the Area Under Receiver Operating Characteristic (AUROC) was .697 (Table 1). Interestingly, excluding MRSA/MSSA status, Agr group and ST/CC had no effect on model performance, suggesting that rare genomic variants are the main driving force of very high δ-toxin production. Recall was poor, suggesting there are yet unfound genomic determinants that contribute strongly to the phenotype. Splitting the toxin levels into 4 categories (0–1000, 1001–7000, 7001–30000, >30000) decreased performance (With metadata: .423 weighted precision, .443 weighted recall, .664 AUROC; Without metadata: .451 weighted precision, .326 weighted recall, .667 AUROC). However, most of the errors in the 4-category model (∼75%) occurred in adjacent categories, suggesting that the classifier was better than random choice with near-misses. ST was the top parameter in prediction, resulting in a loss of ∼21% accuracy when omitted. This suggested that the driving force behind differential δ-toxin level in *S. aureus* is interactions between a potentially large number of genes with the potential to affect toxin expression levels.

## The small chain of carbamoyl phosphate synthase, encoded by *carA*, is necessary for δ-toxin production in USA300

We screened the δ-toxin production phenotype of transposon mutants of 9 of the 42 genes putatively identified by GWAS (Table S3) as well as *agrA* as a positive control and a randomly chosen gene as negative control (SAUSA300_1928). We used mutants from the USA300 Nebraska Transposon Mutant Library (*Fey et al., 2013*) that were transduced

back into the parental USA300 JE2 strains and validated by PCR (Table S2). Of the eleven mutants tested, transposons in *hemL, carA, glpD, isdC, thiD,* and *agrA* significantly reduced δ-toxin production (Fig. 4), but only *carA, agrA,* and *isdC* mutants showed significantly different hemolysis on rabbit blood agar by Mann Whitney U. δ-toxin production in strains containing transposons in *fadD, sbnC, brnQ, hlgB,* and phi77 ORF109-like protein was not different to the parental strain USA300 JE2. *agrA* is necessary to activate RNAIII and *hld* transcription (*Janzon, Löfdahl & Arvidson, 1989*; *Gagnaire et al., 2012*), so the transposon knock-out was expected to completely abrogate expression. However, the complete shutdown of toxin expression in the *carA* mutant had not been previously reported. We showed that δ-toxin accumulation by the *carA* mutant could be rescued by a cloned version of the gene on an expression plasmid (71% δ-toxin production restoration compared to USA300 JE2) but not an empty vector (0% δ-toxin production restoration). *carA* encodes the carbamoyl-phosphate synthase small chain protein, which is involved in L-arginine biosynthesis and UMP biosynthesis (part of pyrimidine metabolism) and has been shown to potentially regulate nitric oxide resistance (*Grosser et al., 2018*) and be important for the regulation of PSMα1 expression (*Hardy et al., 2019*). We confirmed the results of *Bae et al. (2004)* that a *carA* mutant was defective in killing *Caenorhabditis elegans* and showed this phenotype could be restored by complementation. While hemolysis of the complementation strain on sheep blood agar was less than the original strain (Table 2, Figs. 5A–5E), virulence in *C. elegans* was similar. Additionally, complementation with a *carA* allele from a CC45 high δ-toxin strain that restored hemolysis to comparable levels but not δ-toxin production showed reduced killing of *C. elegans* (Fig. 5F). This result suggests that δ-toxin may play an important role in virulence in this model organism.

## DISCUSSION

This study revealed the complex relationship between strain phylogeny and δ-toxin accumulation at stationary phase in *S. aureus*. The phenotype had a strikingly left-skewed distribution, with a minority of strains having >5-fold the median value in toxin units (Fig. 1). Pathoadaptation probably plays a major role in generating this diversity: δ-toxin has been shown to be an important virulence factor in skin and soft tissue infection (*Berlon et al., 2015*), but in bacteremia *agr*-regulated toxins may be under negative selection (*Fowler Jr et al., 2004*; *Sakoulas et al., 2005*). Toxin expression levels may also change through neutral genetic variation or selection on other metabolic pathways, especially if the levels of the toxin are ultimately determined by the interaction of multiple complex regulatory pathways (*Priest et al., 2012*). We showed there was a strong relationship between phylogeny and δ-toxin expression (Fig. 3C). Ancestral reconstruction of δ-toxin levels (Fig. S3) suggested higher expression has evolved several times independently but in a minority of clades, indicative of the fitness trade-offs that can exist with increased virulence. Strains also vary considerably within CCs, suggesting within-clade mutations affect the level of expression. An example of this is the NRS22 strain in CC45, which had a more 4-fold less production than the average for the CC (NRS22 = 6,686; CC45 average = 30,955) (Fig. 2). The association of higher or lower levels with particular CCs is likely due to epistatic interaction
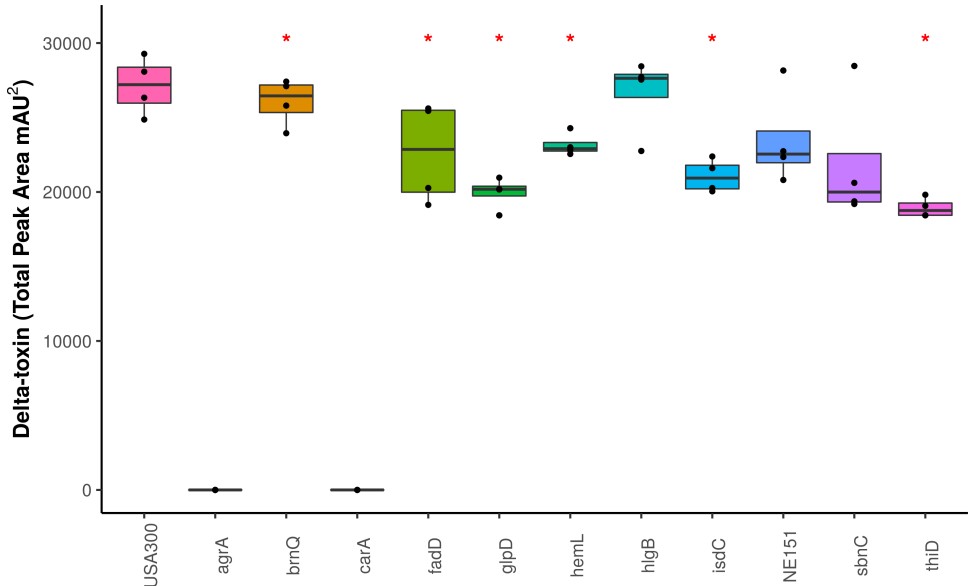

**Figure 4 Impact of gene knockouts on δ-toxin production.** A subset of genes that were found to be significantly associated with δ-toxin by GWAS were tested for their effect of δ-toxin production. δ-toxin from transposon mutants from the Nebraska Transposon Mutant Library (NTML) was measured via HPLC. $*p < .05$.

**Table 2 Characterization of hemolysis activity of USA300, *carA* mutant, and complemented strains.** *S. aureus* strains were spotted on 5% sheeps blood TSA II and incubated at 37 °C for 24 hours followed by 24 hours at 4 °C. Plates were photographed using a lightbox and processed in ImageJ. Hemolysis rings and colonies ($n = 10$) for each strain were measured using ImageJ.

| Strain | Average Hemolysis (mm) | Standard deviation (mm) |
|---|---|---|
| USA300 | 9.0945 | .508 |
| carA-tn | 5.6450 | .360 |
| carA-tn + pOS1 | 5.0459 | .446 |
| carA-tn + pOS1-carA | 7.3305 | .536 |
| carA-tn + pOS1-dtox null carA | 7.0114 | .422 |

between rare mutations and variants shared between clade members. As the strains in this study originate from a wide range of infections (Table S1), it was not possible to associate δ-toxin production in *S. aureus* with a particular disease (such as atopic dermatitis).

We used multiple bacterial GWAS approaches to produce a list of candidate loci that may be affecting δ-toxin production at different phylogenetic levels. GWAS looks for homoplasic genetic variants produced by recombination or parallel evolution that can be associated with phenotypic variation (*Read & Massey, 2014*; *Power, Parkhill & De Oliveira, 2016*). Methods vary by the types of variant tested (SNPs, kmers, indels), whether continuous or discrete phenotypes are used and methods for controlling non-independence of samples due to shared ancestry and typically widespread linkage disequilibrium. Two main approaches have been implemented to determine the underlying population structure of tested bacteria.
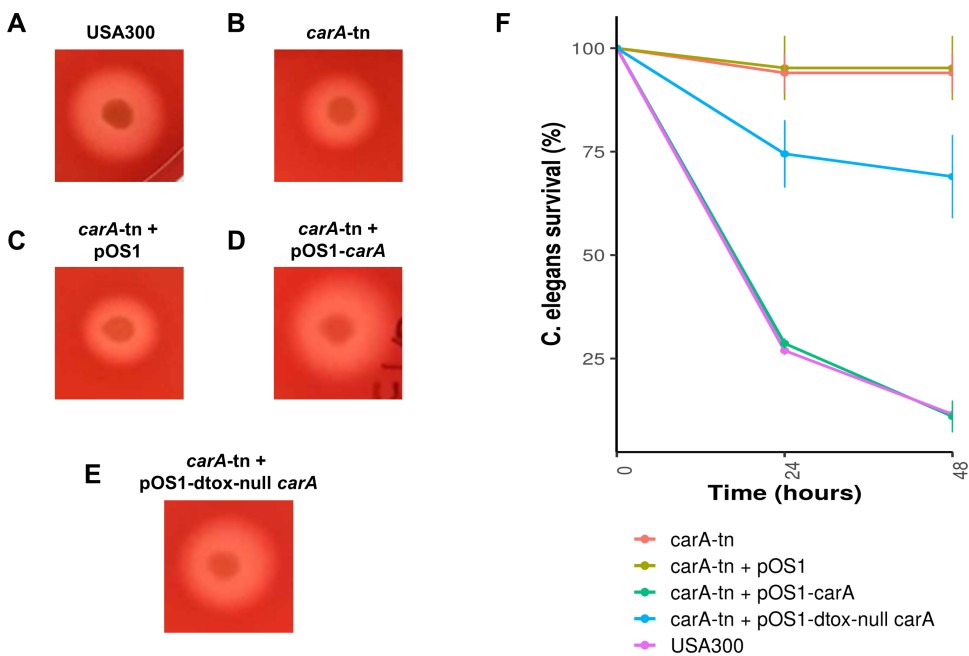

**Figure 5** **Impact of *carA* on hemolysis and virulence of *S. aureus* in *C. elegans*.** (A–E) *S. aureus* strains were spotted on 5% sheep's blood TSA II and incubated at 37 °C for 24 hours followed by 24 hours at 4 °C. Representative colonies are being shown. (F) L4 *C. elegans* were fed either USA300 *S. aureus* (purple), a *carA* transposon mutant USA300 (red), a *carA* transposon mutant with vector (yellow), a *carA* transposon mutant complemented with the native USA300 *carA* allele (green), or a *carA* transposon mutant complemented with a *carA* allele that produces no δ-toxin (blue). Survival was scored at 24 and 48 hours. Survival counts are normalized against C. elegans fed on OP50 *E. coli*.

The most common is a form of principal components correction whereby the genomes of strains are used to create a matrix of linearly uncorrelated variables which can then be included in as either fixed or random effects in a regression model (e.g DBWAS, bugwas, SEER). The second approach uses a phylogenetic tree as input to determine genetic relatedness between strains and can be fed into a regression model as with principal components or used to simulate null data to determine a cut-off for true associations (e.g treeWAS). Although all the variants listed in Table S3 passed the family-wise correction cutoff by their respective programs, many are likely false positives due to the presence of linkage disequilibrium or from underestimation of the underlying population structure, especially since the majority were synonymous substitutions. SEER and treeWAS appeared to be finding SNPs that were common in the *S. aureus* population (in ~1,500–20,000 of the ~44,000 strains in the Staphopia database (*Petit & Read, 2018*)). In contrast, bugwas found SNPs strongly associated with CC45 (the most toxic CC), while DBGWAS found a variant associated with low toxin production that excludes all high toxin strains in CC45. Of note, *mecA* kmers were not discovered by GWAS methods, although phylogenetic regression pointed to MSSA strains having higher toxicity. Similarly, we didn't find any variants within *agr* genes associated with differences in the phenotype, possibly because they were too rare in our population. Although, it is possible to use GWAS to find novel SNPs with

large effect sizes using a relatively small number of genomes within one CC (*Laabei et al., 2014*), the conclusion to be drawn from this pilot study is that larger numbers of *S. aureus* genomes will be needed to understand the δ-toxin phenotype across multiple CCs.

Two other GWAS studies have focused on toxicity in *S. aureus*. Laabei et al were able to build a random forest predictor using 31 SNPs and 21 indels to predict low, medium, and high toxicity in MRSA with an accuracy of >85% (*Laabei et al., 2014*). *Recker et al. (2017)* clarified the role of toxicity by determining factors associated with bacteremia-associated mortality. Five genes, including two in the *agr* operon, were selected by random forest to be predictive of mortality in CC22 and CC30 *S. aureus* bacteremia. None of the genes putatively associated with variation in toxin production from these two studies overlap ours, although our work differed in having a focus specifically on δ-toxin. Genes from our list that have been implicated in other work are *hemL1,* which is part of the *agrA* transcriptional pathway (*Das et al., 2016*; *Young et al., 2017*) and *brnQ*. Other genes found to have reduced rabbit blood hemolysis that did not overlap our set were *hemB, qoxA-C* and *hlgA*. Our GWAS results suggested variants in *qoxD, hemC* and *hlgB* affected in delta toxin production, but these have not been shown to have any toxin-related phenotype in any study we have seen. Mutations in *clpX* and *walK*, which were found in our GWAS results, have also been shown to affect hemolysis (*Frees et al., 2003*; *Delauné et al., 2012*; *Jacquet et al., 2019*).

We validated nine candidate genes for the effect that transposon mutations had on δ-toxin production. Some candidate genes essential for cellular survival (e.g., *clpX* and *walK*) cannot be tested using knockout mutants. Transposons in 5 genes (two predicted by bugwas, two by SEER and one using treeWAS) had no effect on production, indicating they were likely false positive calls. The finding that *hemL, glpD, isdC*, and *thiD* knockouts resulted in a small but significant reduction δ-toxin levels suggests that they may have a significant functional role. The USA300 *carA* knockout had the most dramatic phenotype as the gene was found to be indispensable for δ-toxin production, a result not previously reported. A non-δ-toxin producing *carA* mutant was found to have reduced virulence in *C. elegans,* suggesting a role for δ-toxin in infection. Further mechanistic studies are now needed to understand why *carA* is necessary. Strikingly, the variant discovered through the GWAS screen, and all mutations in the gene in high production CCs such as CC45 were synonymous. There is a growing body of literature documenting differences in protein function caused by synonymous mutations that impact RNA toxicity (*Mittal et al., 2018*) or protein folding (*Walsh, Bowman & Clark, 2019*). Given the involvement of the Agr system, there is possibly a role for mutations to change RNAIII binding specificity and influence gene regulation.

This work (and other GWAS studies) suggest that strain-to-strain variation in δ-toxin production is governed by complex genetic interactions. The high number of significant but probably low effect variants discovered in this analysis highlights the complex regulation of the δ-toxin phenotype and may parallel models proposed for the genetic basis of some traits in eukaryotes (*Boyle, Li & Pritchard, 2017*). Nevertheless, we showed that we can train a classifier (XGBoost) using only genome features with prediction accuracies of 87.9% (binary categories) and 43.5% (4 categories). We found that the most important

predictive feature was ST in the non-binary model, which reflects how much of the variation in δ-toxin production between strains is dependent on phylogeny. The ability to predict phenotypes of toxicity based on sequence data is likely to become an important diagnostic tool as medicine increasingly adopts genome-based technologies (*Laabei et al., 2014*). Going forward, we can improve genome-based predictors and gain mechanistic insights that may lead to development of anti-toxin drugs through a combination of efforts to expand collection of phenotypic variation in natural strains and molecular genetic studies targeted at high-effect loci.

## CONCLUSIONS

δ-toxin production in *S. aureus* is a strain-specific phenotype likely affected by a complex network of genes. GWAS and machine learning approaches have proved successful in determining genetic determinants underlying this phenotype and using them for genome-based prediction. While most genes discovered by GWAS modify δ-toxin production, *carA* was found to be essential. Differences in *carA* function may contribute to virulence by modulating δ-toxin production. Further studies are needed to understand toxin regulation in *S. aureus* and to predict phenotypes from genomic sequences.

## ACKNOWLEDGEMENTS

We would like to thank Levi Morran and McKenna Penley for help with *C. elegans* experiments and John Lees and Ruth Massey for discussion about running GWAS analysis. The discussed strains were provided by the Network on Antimicrobial Resistance in *Staphylococcus aureus* (NARSA) for distribution by BEI Resources, NIAID, NIH: *Staphylococcus aureus*.

### Funding

Timothy D. Read and Michelle Su were supported by the National Institute of Allergy and Infectious Diseases (NIAID) award AI121860. Michelle Su was also supported by the Antimicrobial Resistance and Therapeutic Discovery Training Program funded by NIAID T32 award AI106699-05. There was no additional external funding received for this study. The funders had no role in study design, data collection and analysis, decision to publish, or preparation of the manuscript.

### Grant Disclosures

The following grant information was disclosed by the authors:
National Institute of Allergy and Infectious Diseases (NIAID) award: AI121860.
Antimicrobial Resistance and Therapeutic Discovery Training Program: AI106699-05.

### Competing Interests

Timothy D. Read is an Academic Editor for PeerJ.

## Author Contributions

- Michelle Su conceived and designed the experiments, performed the experiments, analyzed the data, prepared figures and/or tables, authored or reviewed drafts of the paper, and approved the final draft.
- James T. Lyles conceived and designed the experiments, performed the experiments, analyzed the data, authored or reviewed drafts of the paper, and approved the final draft.
- Robert A. Petit III analyzed the data, authored or reviewed drafts of the paper, and approved the final draft.
- Jessica Peterson, Michelle Hargita and Huaqiao Tang performed the experiments, analyzed the data, prepared figures and/or tables, and approved the final draft.
- Claudia Solis-Lemus analyzed the data, prepared figures and/or tables, authored or reviewed drafts of the paper, and approved the final draft.
- Cassandra L. Quave and Timothy D. Read conceived and designed the experiments, authored or reviewed drafts of the paper, and approved the final draft.

## DNA Deposition

The following information was supplied regarding the deposition of DNA sequences:

The sequenced S aureus strains described here are available at NCBI Short Read Archive: PRJNA289526.

## Data Availability

The raw data is available at Figshare: Su, Michelle (2019): GWAS of Delta toxin in *S. aureus* PeerJ #41844. Figshare. Dataset. https://figshare.com/projects/GWAS_of_Delta_toxin_in_S_aureus/69566.

Network on Antimicrobial Resistance in *Staphylococcus aureus* (NARSA) and Nebraska Transposon Mutant Library (NTML) strains were acquired from BEI resources.

## Supplemental Information

Supplemental information for this article can be found online at http://dx.doi.org/10.7717/peerj.8717#supplemental-information.

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
