# Peer review of "Genomic analysis of variability in Delta-toxin levels between Staphylococcus aureus strains"

_PeerJ, doi:10.7717/peerj.8717_

## Round 0.1 · original submission · Major Revisions

Your manuscript has been reviewed by three experts, who all find merit in its content and suggest major revisions of the manuscript be undertaken. If you choose to submit a revised manuscript to PeerJ, please be aware it will be sent out for review.

Reviewer 1 ·

Basic reporting

no comment the manuscript fits the required standards

Experimental design

need improvement (see general comments)

Validity of the findings

Additional work needed to support the conclusion (see general comments)

Additional comments

This study attempts to identify genetic loci potentially associated with variation in δ-toxin production across S. aureus strains. Four GWAS approaches were applied on a set of 124 S.aureus isolates of the NARSA collection. Despite the fact that none of the four GWAS approaches found any significant genetic loci in common, the authors decided to screen the δ-toxin production phenotype of transposon mutants of 9 of the 42 genes identified by GWAS, the rational for this choice being not obvious. The most striking phenotype being that observed with the carA tn mutant, the author focused on this gene. The experiments based on Tn mutant and complemented isogenic strains convincingly demonstrate the carA has a major impact on delta toxin production (I would have looked at RNAIII expression to see wether the effect of carA on agr is trancriptionnal or translational). However, the fact that GWAS identified this gene among other as potential determinant of delta toxin expression is far less convincing because only synonymous mutation in the carA gene were identified. The proposed explanation that such single nt change may have affected RNA stability or RNA binding to a putative unknown target is highly speculative. Only site directed mutagenesis of the carA gene to recreate the single nt mutation observed would be convincing.
Beside, the rest of the paper is interesting, notably the attempt to apply a machine learning approach to predict δ-toxin from genome sequences is promising. However a number of concerns needs to be addressed to validate the model: (i) The splitting of data into training and validation sets with 70/30 proportions is acceptable but leads to high bias in the validation performance estimator. Please consider using 10-fold cross-validation, which should be feasible given the reasonable sample size. (ii) Use of 'accuracy' to report model performance is deceptive here because the binary outcome is not balanced and a 85% validation accuracy can occur by chance. Please provide detailed model performance estimators including ROC-AUC, precision, recall and Cohen's kappa and its associated statistical support. Please only discuss model performance in terms of ROC-AUC and its confidence interval.

Minor comments
line 97. I suggest to reformulate these sentences to better highlight the fact that RNA is a 514 nt regulatory RNA which is the main effector of the agr system. RNA III also encodes the 26 aa delta toxin (which is not the case in every staphylococcus specie), and it is generally admitted that the level of delta-toxin production is a marker of RNAIII expression.

line 112. A specific table showing some detail on the 124 NARSA strains would be appropriate: type of infection, clonal assignment, antibiotic resistance, city or country of origin…

line 150. Please state wether the culture used here was the 15hours agitated TSB described above.

Line 164 : threshold of 20000 ? arbitrary according to Fig S1 ? At least write once in the paper that you have 109 Low versus 15 High -producing strains

line 167. Hemolysis assay. Why not using the synergistic hemolysis assay (cross-streaking with a strong beta-hemolysis producer such as RN4220) which is slightly more specific for delta toxin than the crude hemolysis diameter?

line 182. Concerning SEER GWAS method: « Dimensions were chosen… » How many dimensions ? and is it possible to show the plot in Supplementary data ?

Line 219 : 14 predictors = CCs + MRSA + Agr + 11 predictors from Figure 4 ?

Line 226 & 283. Two different Pagel lambda values for the phylogenetic signal of d-toxin production are given (lambda = 0.50 line 226, = 0.99 line 282). Please clarify the differences between these models.

Line 238. Statistics. Did you perform multiple test corrections for the pairwise Mann-Whitney U tests?

line 281 and Fig.2. Some red barrs that represent delta-toxin assessment on fig2 are apparently missing for some CC (ie CC 8). Is it because the production was undetectable ?

Line 283 regarding Blomberg K and Pagel lambda, "for both measures, a value of 1...": this sentence is unclear given that K << 1 and lambda ~ 1, please detail the interpretation (the current sentence reads as if Blomberg model contradicted the Lambda model)​

line 291. I suggest to illustrate "high δ-toxin producing clades » by adding: "such as CC45, CC890 or CC72.."

line 317. "87 strains in the “low” toxin and 19 in the high toxin category ». This totalizes 106 strains; where are the remaining 18 strains ?

line 376. " δ-toxin accumulation by the carA mutant could be rescued by a cloned version of the gene on an expression plasmid (but not the empty vector) » : it would be nice to include the rescued strain in Fig 4.

line 385. « complementation with a different carA allele » What is this allele ? how was it chosen ?….. the rational for these experiment is not clear.

Of note I could not find the figure legend of Fig.S1 and S2

·

Basic reporting

I believe the article meets your standards

Experimental design

• Why did you choose to look at the delta toxin?
• How did you choose the 124 strains that you examined, which infectious syndromes were they from (was this a single infectious syndrome or many different ones?)
• Was there a reason to believe that many of the strains would be delta toxin positive – did you perform some statistical analysis before ordering the strains to determine how many needed to be delta toxin positive and delta toxin negative? What was the statistical plan for determining the sample size?
• Does 124 strains provide enough strains given that you were analysing the whole genome?
• I would suggest that a proper case control study needs to be performed where you examine strains causing only atopic dermatitis compared with stains carried in the anterior nares to understand any association between delta toxin and atopic dermatitis, the numbers for each group needs to be determined using power calculations.

Validity of the findings

• It’s interesting that none of the four GWAS approaches found any significant genetic loci in common, why do you think this is?

Toxin identification using HPLC:
- why did you use an arbitrary cutoff of 20,000? Is there supporting evidence for this cutoff?

GWAS:
- Why did you chose a MAF of 0.2 because of the low sample size, please explain further
- What was the reasoning behind using strain N315 rather than other strains?

Phylogenetic Regression:
- Explain further why you assumed a Brownian Model for the evolution of  toxin given the estimate did not entirely fit under the BM assumption

Additional comments

Additional comments are listed below:

Discussion:
- I’m interested to understand why the variants that passed the family-wise correction cutoff were thought to be likely false positives
- Why did the DBGWAS variants exclude CC45

Figure 1 – there are only a few strains with >30,000 delta toxin on the x axis – these are interesting, which strains are these and what are the units?

Reviewer 3 ·

Basic reporting

No comment.

Experimental design

The authors do define the research question well, but the methods require additional detail in several key places to allow for replication.

Validity of the findings

Not all underlying data is provided. The authors should link the sequence data to the delta-toxin production levels.

Additional comments

The authors present an interesting manuscript, performing multiple GWAS analyses to attempt to elucidate the genetic loci involved in delta toxin production in S. aureus. However, the paper appears to lack detail in several key areas. In addition, some of the analyses involve the selection of arbitrary cutoff values, that will require either further justification, or investigation into the effect of adjusting these values. Lastly, while the authors do publicly release the sequence data underpinning the paper, there appears to be no way to link this back to the toxin-production values, and so no way to verify or repeat the authors’ analysis.

Major Comments
Line 121: “Whole-genome shotgun sequencing”. This section requires more detail. The authors state Illumina chemistry was used, was this v2 or v3? Paired-end sequencing was performed, but what length reads? Was single or dual indexing performed?

Line 124: “Raw read data…” Were a minimum number of reads required for each strain, to be included in the study? If so, were additional sequencing runs performed? Or were strains simply excluded? Some of the strains listed in the SRA project accession number have a very low number of bases, e.g. NRS109 has only 12.7M bases (spread across two sequencing runs), NRS168 has only 7.7M bases (spread across two sequencing runs), while others have upwards of 400M bases. Also, none of the strains listed under this SRA project accession number are stated to have been sequenced using Illumina HiSeq (all are MiSeq), but the authors state both MiSeq and HiSeq was used.

Lines 125-126: “data were deposited in the NCBI Short Read Archive under project accession no. PRJNA289526.0”. Was unable to access this to verify (can find them without the “.0”). Does the publication or SRA project include any information that would link the raw sequence data to either the binary high/low phenotypes of toxin production, or the actual toxin production levels, so that others can attempt to replicate the analysis? I couldn’t find that information anywhere. This would be an important point to allow replication of experiments – just release of sequencing data is not sufficient.

Line 130: “Read trimming to eliminate Illumina adapters and for quality”. What parameters were used for trimming based on quality?

Lines 130-131: “Read errors and de novo genome assembly was performed”. What does read errors was performed mean? Were any parameters required to be met to keep an assembled genome (minimum N50? No. of contigs? Size of genome?).

Line 164: The authors used an “arbritary cutoff of 20,000” for determining binary Low/High of toxin production. What effect does varying this value have? For example, what if the median value of 8295 units was used?

Lines 193-194: “treeWAS was run with three unrooted trees”. But two of the trees the authors then describe are, by their own description, rooted.

Lines 235-236: “Strains used in the GWAS are split among the training and validation set”. Beyond the 70%/30% split, how was this performed? Randomly? Did you force each set to contain a certain number of low/high toxin producers? Did each set contain representation across the clonal complexes, or at least the two main S. aureus groups?

Lines 332-333: “Using a ranked phenotype, genes common to all three analyses were fadD, vraD, degA, gdpP, ggt, opcR, rebM, thiD, and three uncharacterized proteins”. Table S1 also lists sufB for treeWAS? Was this not common to all three analyses? gdpP, opcR and rebM are not listed in Table S1. Do these only have the protein accession number/description? They should be labelled if being referred to specifically.

Minor Comments
Line 55: “CC30 had significantly lower levels of and toxin production”. Do you need the ‘and’ there?

Lines 134-135: “Agr type was determined using BLAST”. Using nucleotide or protein sequences? What coverage and identity cutoff values were used?

Line 143: “and to obtain a downsized core genome alignment of 42,406 bp” Is this just SNPs, or does it contain invariant data also (checking as you don’t include ‘I’ in the GTRGAMMA model for RAxML). If it does include invariant data, how many SNPs were there? Were gaps allowed in the alignment?

Line 144: What version of RAxML?

Lines 250-251: “All transposon mutants were… confirmed by PCR”. It might be nice to include a table of the primers in the supplementary data.

Lines 252-253: “Complementation was performed by cloning PCR fragments containing the USA300 genes in to the pOS1-Plgt vector and transforming back into the mutant strains”. Again, more detail is required here. Primers for PCR fragments? Conditions for transformation?

Line 296: Agr types: “NA:3”. The authors could comment on these. What does N/A mean here? Was agrD missing? Hybrid? Novel? If missing, was it because it was at the end of a contig? Or no BLAST match due to not meeting cutoff parameters for identity?

Line 297: Kruskal-Wallis, not “Kruskal-Wllis”.

Line 316-316: “which gave a set of 87 strains in the “low” toxin and 19 in the high toxin category”. That’s only 106 strains? There were 124 in total (although line 141 states 125 strains), so what happened to the other 18?

Line 369: “Of the eleven mutants tested, transposons in hemL, carA, glpD, isdC, thiD, and agrA significantly reduced δ-toxin production”. Discounting the positive control (agrA), the remaining five genes are fairly evenly split from across the GWAS methods (two detected using BUGWAS, two detected using Seer and one detected using TreeWAS). Might be worth commenting on in the discussion?

Lines 402-403: “while CC45 strains, which are high producers are associated with moderately severe disease”. It was unclear as to whether the authors meant severe disease in general, or for cases of AD.

---

## Round 0.2 · Minor Revisions

Thank you for submitting your revised manuscript. As you will see, one of the reviewers has asked for minor revisions to be undertaken. Please address these and submit your revised manuscript at your earliest convenience.

·

Basic reporting

No comment

Experimental design

No comment

Validity of the findings

No comment

Additional comments

The manuscript describes an analysis of delta-toxin levels in different Staphylococcus aureus strains isolated from 9 countries and a number of different infections between 1935 and 2003 requested from the Network on Antimicrobial Resistance in Staphylococcus aureus (NARSA) biobank. The group studied the delta toxin of S. aureus because it has been shown to be linked with atopic dermatitis and they postulated that there would be a variation in toxin production from different S. aureus strains. The group examined a set of 124 S. aureus strains which were examined using whole genome sequencing and analysed using a number of genome wide association studies for association with high levels of production and specific genes. A very diverse set of samples were examined, the multiple approaches suggested the potential association between toxicity and 45 different genes, this was examined further with machine learning models and one gene (carA) was examined further and suggested to be involved with toxin production and toxicity for Caenorhabditis elegans.

Thank you for answering the previous comments, please find some further questions below.

Questions:
• I would like to see a paragraph at the end of the introduction outlining what the study aims to achieve as an outline before the materials and methods
• There needs to be more explanation about the strains that you studied. As I understand, you are interested in inflammatory skin diseases such as atopic dermatitis. The strain collection you examined were very mixed in the infections they caused – ranging from bacteraemia to impetigo and surgical incision.
• With strains from such a mixed clinical background (with some unknown) I wonder if a GWAS approach on these samples is at all likely to find associations, compared to examining one clinical infection. This needs to be outlined in your discussion where a mixture of different clinical presentations might confound your analysis and dilute any associations with skin infections.
• The limitations of the study need to be outlined fully in your discussion


Minor comments:
• Conclusion section in first abstract: “The amount of stationary phase production of the toxin is a strain-specific phenotype likely affected by a complex interaction of number of genes with different levels of effect” should this read “… of a number of genes with different levels of effect’
• The wording between the first and second abstracts are not the same, for example the results in the first abstract states “… with a precision of .875 but recall of .333” and the results section in the second abstract states “with a precision of .875 and specificity of .990 but recall of .333”; please ensure that the wording is the same for both

Reviewer 3 ·

Basic reporting

No comment.

Experimental design

No comment

Validity of the findings

No comment

Additional comments

The authors present an improved version of the manuscript, having addressed all of the reviewers’ comments. I have no concerns about the new version.

---

## Round 0.3 · accepted · Accept

Thank you for addressing Dr Moore's request for minor revisions. I am delighted to inform you that your manuscript has been accepted. Thank you for taking on board and addressing all reviewers' comments during the review process.